# The Translational Role of Animal Models for Estrogen-Related Functional Bladder Outlet Obstruction and Prostatic Inflammation

**DOI:** 10.3390/vetsci9020060

**Published:** 2022-01-31

**Authors:** Risto Santti, Emrah Yatkin, Jenni Bernoulli, Tomi Streng

**Affiliations:** 1Institute of Biomedicine, University of Turku, FI-20014 Turku, Finland; emryat@gmail.com (R.S.); jennibernoulli@gmail.com (J.B.); 2Central Animal Laboratory, University of Turku, FI-20014 Turku, Finland; emryat@utu.fi; 3Laboratory of Animal Physiology, Department of Biology, Division of Genetics and Physiology, University of Turku, FI-20014 Turku, Finland

**Keywords:** prostatic inflammation, lower urinary tract symptoms, bladder outlet obstruction, animal models, estradiol to testosterone ratio

## Abstract

The prevalence of LUTS and prostatic diseases increases with age both in humans and companion animals, suggesting that a common underlying cause of these conditions may be age-associated alterations in the balance of sex hormones. The symptoms are present with different and variable micturition dysfunctions and can be assigned to different clinical conditions including bladder outlet obstruction (BOO). LUTS may also be linked to chronic non-bacterial prostatitis/chronic pelvic pain syndrome (CP/CPPS), but the relationship between these conditions is unknown. This review summarizes the preclinical data that supports a role for excessive estrogen action in the development of obstructive voiding and nonbacterial prostatic inflammation. Preclinical studies that are emphasized in this review have unequivocally indicated that estrogens can induce functional and structural changes resembling those seen in human diseases. Recognizing excessive estrogen action as a possible hormonal basis for the effects observed at multiple sites in the LUT may inspire the development of innovative treatment options for human and animal patients with LUTS associated with functional BOO and CP/CPPS.

## 1. Lower Urinary Tract Symptoms and Chronic Prostatic Inflammation

### 1.1. Lower Urinary Tract Symptoms

The prevalence of lower urinary tract symptoms (LUTS) increases with age [1,2]. Conditions leading to LUTS are life-long processes that become more common between the age of 20 and 49 years [3,4]. LUTS in men are usually divided into storage symptoms (e.g., increased daytime frequency and nocturia), voiding symptoms (e.g., slow or intermittent stream, terminal dribbling), and post micturition symptoms (e.g., feeling of incomplete emptying of the bladder) [5,6]). It has been established that prostate growth continues throughout life, and the very first pathological signs of benign prostatic hyperplasia (BPH) appear in adult males before 30 years of age [7], eventually leading to bladder outlet obstruction (BOO) along with a continuous decline in the concentration of total testosterone in the serum [8]. BPH or prostate enlargement are also age-related in dogs with similar age-related hormonal changes as seen in humans [9,10]. Especially acute and chronic prostatitis seems to be associated with bacterial infections in animals, but the role of non-bacterial inflammation in LUTS is poorly understood. In cats, prostatic disorders are extremely rare; nevertheless, feline lower urinary tract disease (FLUTD) describes any obstructive or non-obstructive disorder affecting the urinary bladder or urethra [11].

The incidence of severe urinary symptoms is higher in Japanese men without BPH [12] and the flow rate is comparable between Japanese and American men with enlarged prostates. It has also been shown that improvements in uroflowmetry do not correlate with a reduction in symptoms [13]. On the other hand, asymptomatic patients may display obstructive voiding [14] while symptomatic patients may display non-obstructive voiding [15,16].

In a study by Bosch et al. [17], the variation in prostate volume was found to determine variation in urethral resistance by 15% or less depending on the parameters used. They suggested that the different pathophysiological mechanisms that can increase urethral resistance in patients with BPH are mainly determined by factors other than the volume of the prostate. Mattiasson [18] suggested that other parts of lower urinary tract (LUT), along with the prostate, should be taken into account when treating patients with LUTS. In fact, no clinically relevant relationships have been found between total symptom score, prostate size and average urine flow [19,20]. The association between smaller prostates and the frequency of BOO suggests that early male estrogenization may be involved [21].

The physiological function of the bladder outlet is complex and may be crucial in regard to the mechanisms underlying LUTS [22]. BOO, in its functional condition, has been suggested as a cause when the bladder outlet does not open appropriately during voiding [22,23,24]). Functional BOO may refer to bladder neck [25], external striated urethral sphincter (EUS) [26], or rhabdosphincter (RB) [27] dysfunction. Taken together, the muscular layers of the LUT are commonly accepted to play an important role in the development of increased urethral muscle tone or, when dysfunctional, in inappropriate relaxation.

It is important to remember that diagnosis of BOO is dependent on measurements of pressure and flow rate [28]. The conditions mentioned above can together be defined as BOO, which is the generic term for LUT obstruction [29]. The purpose of this review is to summarize the functional causes of BOO.

### 1.2. Anatomical and Functional Aspects of LUT Sphincters

In spite of studies involving electromyography (EMG) [30], videourodynamics [31] and ultrasonography [32], relatively little is known about the natural history, incidence, and course of functional BOO at the upper urethral level. However, the role of all urethral sphincters may be central to the development of voiding dysfunctions. Traditionally urethra-related dysfunction has focused on the EUS caused by trauma to the upper spinal cord, transverse myelitis, multiple sclerosis, and myelodysplasia [33,34,35]. There may also be non-neurogenic causes of external sphincter dyssynergia [4]. Similar causes are associated with urethral sphincter mechanism incompetence in animals, especially in dogs [11]. Urodynamic studies in anesthetized dogs or in conscious dogs with telemetry has been recognized and used in veterinary practice [36,37].

By definition, dyssynergia refers to inappropriate contraction of the urinary sphincter(s) during voiding or failure of complete relaxation. Bladder neck dyssynergia has been suggested to be a cause of functional voiding dysfunction [38,39,40,41]. In dyssynergia of the bladder neck area, the musculature of the site actively tightens during voiding. Less is known about smooth muscle dysfunctions, but detrusor-urethral sphincter dyssynergia (DUSD) is commonly observed and is known to cause functional obstructive voiding [30,33,42]. It has been known for decades that DUSD can cause urinary retention [26]. Urethral sphincters play a central role in micturition and the development of voiding dysfunctions. The complex structures and functions of the sphincters and their dysfunctions have been widely discussed in the book named “The Urethral Sphincter” [43]. There are regional and age-related variations in urethral smooth muscle in addition to striated muscles. It is likely that the structural changes lead to altered function, which may contribute to the development of obstructive voiding.

### 1.3. Prostatic Inflammation and LUTS

In addition to micturition dysfunctions, LUTS may be linked to chronic nonbacterial prostatitis/chronic pelvic pain syndrome (CP/CPPS). CP/CPPS can present with a range of clinical manifestations, the main symptoms of which include urogenital pain and LUTS with voiding or storage symptoms [44,45].

The histological term prostatitis is used to describe the presence of acute and chronic inflammatory cells in the prostate and is a common histopathological observation. A standardized histopathological system for chronic prostatic inflammation was proposed by Nickel et al. [46], in which prostatic inflammation is classified according to its extent (focal, multifocal, diffuse) and grade/severity (mild, moderate, severe) in each anatomical location (glandular, periglandular, and stromal). Modification of this histological classification has been used, e.g., to include quantification of acute and chronic inflammation [47]. However, it is important to note that the diagnosis of CP/CPPS is usually based on symptoms and clinical signs with or without evidence of inflammatory cells in the prostatic tissue or secretions. The association between histological inflammation and CP/CPPS has remained uncertain. Patients with chronic prostatitis may be more likely to develop BOO or urinary retention than men without prostatic inflammation. The relationship between histological inflammation and symptoms of prostatitis and LUTS was extensively studied in the REDUCE trial, which concluded that inflammation is not associated with a greater risk of CP/CPPS-like symptoms, but chronic inflammation nonetheless predicts symptom progression in men suffering with CP/CPPS [47,48,49]. Physician-diagnosed prostatitis may therefore be an early marker or a risk factor for the development of later prostatic or urological problems [50,51].

Another potential pathophysiological mechanism in patients with CP/CPPS is the inability to voluntarily relax the rhabdosphincter/external urethral sphincter (RB/EUS), as mentioned above. This can result in dysfunctional high-pressure voiding and intraprostatic ductal reflux, which is thought to be implicated in the pathogenesis of CP/CPPS. Mehik et al. [52] have shown an association between elevated prostate tissue pressure and inflammation. It is also possible that inflammatory cells secrete cytokines that can exert their effects locally on smooth and striated muscular fibers embedded in the glandular tissue that are anchored to structures connecting the prostate to the bladder and pelvic floor. Thus, the excessive activity of periurethral and periductal smooth muscle cells may also contribute to the development of nonbacterial prostatitis and voiding dysfunctions.

## 2. Sex Hormone Imbalance as a Cause of LUTS and CP/CPPS

There are several lines of evidence indicating that the altered action of sex hormones, particularly the imbalance between estrogen and androgen serum concentrations, may be a common cause of a broad range of clinical conditions. Estrogens and androgens target estrogen receptors (ER) and the androgen receptor (AR), respectively. Both receptor types have been well characterized in the prostate, but their roles in human male LUT are poorly understood. In the adult male rat, co-expression of AR and ER-beta has been reported in the urothelium, bladder smooth muscle cells, proximal urethra and prostatic autonomic ganglia of the prostatic plexus suggesting that the hormones have a direct effect on LUT manifestation [53,54]. A bladder biopsy study in humans indicated that ERs are expressed in the bladder and concluded that ER-alpha is the key mediator in the various symptoms associated with BPH/LUTS [55].

It is important to recognize that exposure to estrogens and a gradual alteration in hormonal ratio is a life-long process in men that may be caused at different ages by endogenous or environmental factors [56]. Excessive estrogen exposure in utero or during early life, referred to as estrogen imprinting or developmental estrogenization, can lead to increased disease risk with aging as we have shown in various experimental models [57,58,59,60]. This phenomenon has recently been reviewed comprehensively by Prins [61]. During aging, the concentration of total testosterone, and perhaps more importantly its biologically active portion, progressively declines in serum in men [8], while aromatase activity (required for conversion of testosterone to estradiol) increases with advancing age or increasing fat mass [62]. The influence of age on estradiol levels is less clear, however it has been reported that total serum estradiol increases with age among men [63]. In opposition to this, another study did not find evidence that estradiol concentration is associated with age but instead that estradiol change has a strong direct association with changes in testosterone levels [62]. This highlights that estradiol levels may be predominantly dependent on testosterone and thus that diminishing testosterone levels while aging is the primary determinant of changes in estradiol concentration [62]. Moreover, an age-related increase in sex hormone-binding globulin [8], decreases the bioavailability of testosterone to an even greater extent, which conceivably leads to amplified estrogen action.

The age-related increase in the ratio of estradiol to testosterone in serum hints at a possible link to LUTS and CP/CPPS. Regarding prostate size, two cross-sectional studies have shown a significant positive correlation between serum estradiol and prostate size, and between total estradiol (together with the ratios of estradiol to testosterone and estradiol to free testosterone) and LUTS in symptomatic men aged 60 years or older [62,64,65]. An age-dependent increase in the severity of LUTS has been reported along with higher plasma estradiol which is associated with an increase in storage and voiding symptoms [2]. The conclusions drawn from these findings and those of Chavalmane et al. [66] support the theory that increased estradiol and an increased estradiol to testosterone ratio in serum are important factors in LUTS. The theory that hormonal imbalance during aging is associated with prostatic inflammation is not however fully accepted. In one study, some hormonal abnormalities in men with CP/CPPS were identified but no difference was found in estradiol concentrations as compared to the control group [67]. On the other hand, it has been suggested that some genetically predisposed CP/CPPS patients may be less responsive to the protective effects of testosterone in the stroma, which thereby leads to an accumulation of estradiol and consequentially inflammation [68]. It was shown that the combined treatment with androgens and estrogens induced prostatic enlargement and multifocal inflammation in dogs [69,70]. Prostatic inflammation was considered to be the consequence of morphological alterations within the prostate that is undergoing hyperplasia as a result of a hormonal imbalance [70].

Owing to the increasing prevalence of LUTS and functional BOO, and CP/CPPS with age, there may be some dependence of these phenomena on the age-related decline in testosterone concentration and concurrent elevation of estrogens. However, there is evidence suggesting that testosterone replacement therapy may worsen LUTS and that hypogonadism is an important risk factor for LUTS/BPH [71]. Efforts to identify testosterone deficiencies have not considered the physiological consequences of various testosterone serum levels, nor the age-associated variation when the diagnosis of testosterone deficiency is made. Thus, the possible mechanisms by which testosterone and/or estrogen or their ratio can influence the development of LUTS and CP/CPPS in men have not been fully elucidated.

## 3. Preclinical Models for Estrogen-Related Lower Urinary Tract Functions

In order to deepen the understanding of relationships between altered hormonal ratio and functional BOO and CP/CPPS, our research team has focused on experimental translational studies for decades. Our aim has been to reveal sites of estrogen action in the male LUT and prostate, and to establish models for functional BOO and prostatic inflammation, in order to determine whether estrogen-directed treatments are therapeutically beneficial in these experimental models. Studies in these experimental disease models have indicated that estrogens induce both structural and functional changes resembling those seen in human diseases such as BOO and CP/CPPS.

Herein we introduce an array of animal models [57,60,72,73,74,75,76,77,78,79,80,81,82,83] that have been generated by our research group to recapitulate functional BOO and histological prostatic inflammation as induced in humans with hormone imbalances (Figure 1). Despite anatomical differences, the rodent LUT shares many similarities with the human LUT (Figure 2 and Figure 3). Our group has introduced a novel method for the recording of EMG activity of the urethral musculature combined with histology [72], transvesical cystometry, and flow rate in rats [73] and models in mice [74,75,76], in order to analyze the role of androgens and estrogens in the development of functional BOO. Hormonal imbalance may be caused by several different hormonal exposure approaches during the neonatal period or in adult mice and rats. Detailed experimental methods have not been described in this review, but it is essential to understand that mice and rats, as well as different strains, differ in their sensitivity to hormone imbalance. Therefore, it is always important to follow carefully used exposure methods in the original publications.

### 3.1. Male Estrogen-Related BOO in Rodent Models

We have established various disease models that aim to increase our understanding of the role of estrogens, and especially an imbalance in the ratio of estrogen to androgen in serum, in causing voiding dysfunctions. Neonatal estrogenization of male rats by diethylstilbestrol has been shown to cause non-traumatic infravesical obstruction. The obstruction is associated with rhabdosphincter atrophy, due to urethral opening failure caused by reduced transient repolarizations that lead to failure in the function of the muscle—functional BOO [60]. This rat model has been used for testing the therapeutic potential of aromatase inhibitors in treating functional BOO [59], where the inhibitors had a positive therapeutic effect. A transgenic mouse strain expressing human P450 aromatase gene under the ubiquitin C promoter (AROM+) also displays functional urethral obstruction associated with rhabdosphincter atrophy [75]. Since it is known that estrogen exposure during pregnancy may cause developmental abnormalities [77], AROM+ male mice is considered as a representative model for early estrogenization. In a study by Li et al. [78], aromatase inhibitors (finrozole and letrozole) were shown to ameliorate symptoms in these AROM+ male mice.

The findings in these developmentally estrogenized rodent models suggest that endogenous estrogens (or androgen deficiency) may cause functional RB disorder and that this could potentially be treated with aromatase inhibitors. Endogenous estrogens may also be involved in the etiopathogenesis of non-traumatic DUSD in men. The effect of aromatase inhibitors on non-traumatic obstruction in men remains to be shown.

### 3.2. Prostatic Inflammation and Obstructive Voiding in Rodent Models

Estrogen-induced prostatic inflammation is a well-described phenomenon in the adult rat and mouse [84,85,86]. However, previous animal studies did not explore the impact of prostatic inflammation on voiding-related dysfunctions and symptoms. Therefore, our aim was to explore the role of estradiol and testosterone in the development of prostatic inflammation and voiding dysfunction by establishing an experimental model combining these aspects. Although neonatal estrogenization in mice causes obstruction, it has also been observed to inhibit growth of the prostate lobes and cause severe inflammation [57]. Thus, neonatal estrogenization was not considered as appropriate treatment method for generating a representative model for prostatic inflammation studies in adult mice.

The novel animal model of obstructive voiding and nonbacterial prostatic inflammation was based on the exposure of adult non-castrated Noble rats to excessive amounts of estradiol in the presence of physiological amounts of androgens. Histopathological studies were performed on the dorsolateral prostatic lobes (DLP) that closely represent the corresponding sites in the human prostate (Figure 3). Accumulation of lymphocytes in the perivascular space in the DLP after 3 weeks of treatment was the earliest sign of estrogen action [79]. This finding suggests that an estrogen-induced increase in vascular permeability may be an important step in the inflammatory response. When the hormonal treatment period was extended to 6 weeks, inflammation proceeded to the stroma, and finally, after 13 weeks on hormonal treatment, the inflammation took a complex glandular form [80]. Glandular inflammation mostly took the form of neutrophil infiltration into the glands, which is characteristic for the acute phase of chronic inflammation. To assess inflammation, a histopathological counting system was established based on classification of human prostate inflammation sections [46]. Inflammation infiltrate area, extent and grade in the DLP were assessed in different anatomical locations (perivascular, stromal, glandular). Our later studies showed that (1) inflammation and epithelial alterations were impacted by the androgen to estrogen ratio [81], (2) glandular inflammation could be attenuated by treatment with the selective estrogen receptor modulator (SERM) fispemifene or dietary soy [82,83], and (3) that inflammation in the DLP did not develop into adenocarcinomas, which were observed in the periurethral region after 18 weeks of hormonal treatment [87].

Urodynamical measurements (micturition cycle, bladder volume and volume of residual fluid, flow rates, bladder pressures) were performed concomitantly in intact male Noble rats treated with estrogen and androgen. After 3 weeks of hormonal treatment, when the first signs of perivascular prostatic inflammation were seen, incipient signs of altered bladder function were also observed [79]. Prolonged hormonal treatment for 13 weeks induced more extensive prostatic inflammation and urethral obstruction associated with rhabdosphincter dysfunction [80]. The ratio of testosterone to estradiol was shown to be essential in voiding dysfunction as no obstructive changes were observed in hypoandrogenic and hyperestrogenic animals although prostatic inflammation was prominent. When the testosterone concentration was increased above control levels, the animals developed both obstructive voiding and glandular inflammation [80]. These diverse responses do not support the hypothesis that nonbacterial prostatic inflammation is sufficient for the development of obstructive voiding. Prostatic inflammation may be a disease on its own and voiding alterations may represent separate non-prostatic estrogenic effects.

## 4. Concluding Remarks

As there is a high number of men experiencing functional BOO/CP/CPPS, which greatly affects the quality of life, the use of animal models is essential to explore possible medical treatments. In this review, we have shown evidence indicating that estrogens and altered sex hormone action, particularly imbalance between the actions of estrogen and androgen, may be a common cause of broad range of clinical conditions. The age-related increase in the ratio of estradiol to testosterone in serum coincides with an increase in the prevalence of functional BOO/CP/CPPS. LUTS and its clinical manifestation, BOO, may be caused by hormonal or anatomical dysfunction. The causal relationship between functional BOO and CP/CPPS remains to be further explored.

There are obvious differences between human and animal anatomical structures and functions. Nonetheless, animal models play an important role in our attempts to understand and mimic human conditions. Prof. Andersson and colleagues have stated that “the use of animals is needed for understanding pathophysiological mechanisms involved in the human male e.g., bladder neck obstruction” [88]. It is challenging to interpret data from animal studies and translate these findings to human conditions. In spite of this, in our studies we have categorically established various models to explore simultaneous human functional BOO and prostatic inflammation caused by altered estrogen and androgen action. Neonatally estrogenized rodents, aromatase overexpressing mice and adult rats exposed to altered ratios of estrogen to androgen have provided important insights into the links between hormonal dysfunction, BOO and prostatic inflammation, although it is difficult to discern the causal relationship between them.

Finally, it is justified to ask whether estrogens contribute to the development and persistence of prostatic inflammation and functional BOO. Anatomical and functional sites in the LUT suggest that there are multiple therapeutic targets for modulating excessive estrogen action. Additional research is needed for understanding both the limitations and possibilities of animal models for functional BOO, and CP/CPPS may provide possibilities for translatability and developing novel therapeutic modalities. Therefore, better understanding of lower urinary tract disorders using laboratory animals as disease models will ultimately benefit not only humans but also animals.

## Figures and Tables

**Figure 1 vetsci-09-00060-f001:**
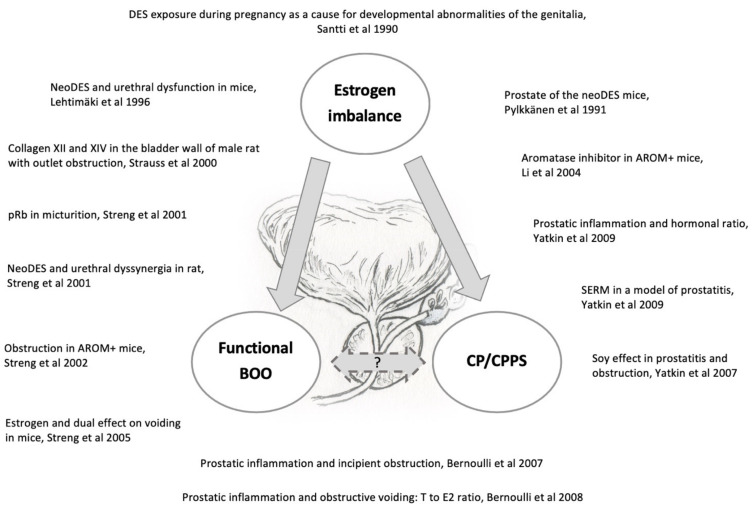
Imbalanced estrogen action causes both structural (e.g., prostatic inflammation) and functional (e.g., bladder outlet obstruction) dysfunction in the lower urinary tract as depicted in our several preclinical rodent studies [57,60,72,73,74,75,76,77,78,79,80,81,82,83].

**Figure 2 vetsci-09-00060-f002:**
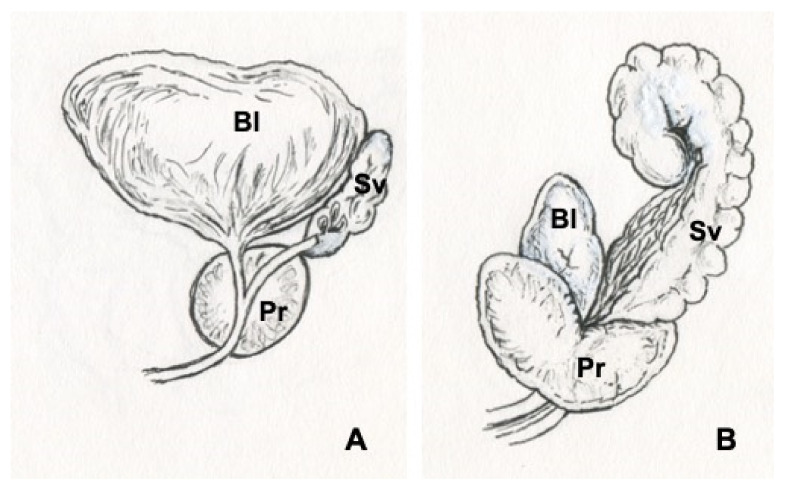
Illustrations of human (**A**) and rat (**B**) prostates. Bladder (Bl), prostate (Pr), seminal vesicles (Sv).

**Figure 3 vetsci-09-00060-f003:**
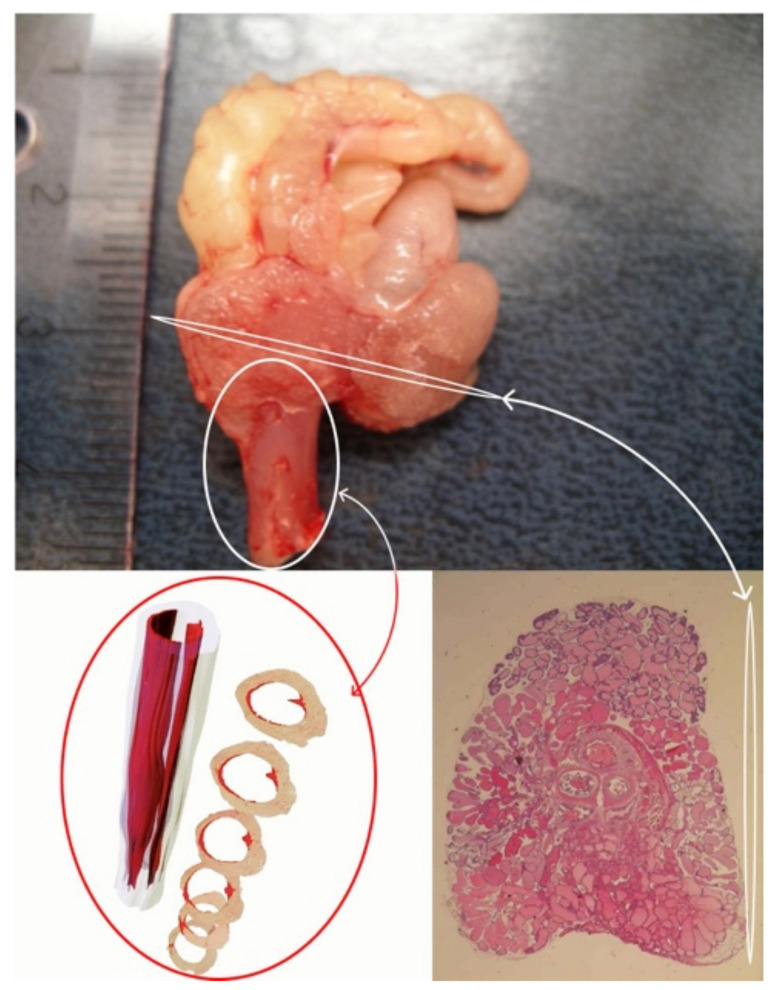
Transverse section of a rat prostate and 3D model of a rat rhabdosphincter.

## Data Availability

Not applicable.

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
