# Peer review of "The Translational Role of Animal Models for Estrogen-Related Functional Bladder Outlet Obstruction and Prostatic Inflammation"

_vetsci, 2022, doi:10.3390/vetsci9020060_

Round 1
Reviewer 1 Report
Aim of the present manuscript is to summarize the potential role of animal models for the understanding of the pathogenesis of BPH and LUTS/BPH in human subjects.
The review is comprehensive and available evidences are clearly reported.
Topic is current and relevant due to the large prevalence of BPH in adult subjects; therefore, a deep understanding of limitations and potentials of animal models for LUTS may provide possibilities for developing novel therapeutic modalities ultimately benefiting both humans and animals.
Manuscript is well written and Figures are clear.
Author Response
Reviewer 1
Comments and Suggestions for Authors
Aim of the present manuscript is to summarize the potential role of animal models for the understanding of the pathogenesis of BPH and LUTS/BPH in human subjects.
The review is comprehensive and available evidences are clearly reported.
Topic is current and relevant due to the large prevalence of BPH in adult subjects; therefore, a deep understanding of limitations and potentials of animal models for LUTS may provide possibilities for developing novel therapeutic modalities ultimately benefiting both humans and animals.
Manuscript is well written and Figures are clear.
Reply: We deeply thank for this valuable response.
Reviewer 2 Report
Thank you very much for giving me this opportunity to review the article entitled "The translational role of animal models for estrogen-related functional bladder outlet obstruction and prostatic inflammation."
The authors overviewed sex hormone-related LUTS and its animal models clearly and concisely; however, some minor modifications are required.
The followings are comments and questions.
-Third part, Preclinical models for ---, seems to be the main point of this article. For the better understanding by readers who are interested in LUTS and animal experiments, it would be better to have a more detailed explanation. Some questions are listed below.
-Neonatal estrogenization method and AROM transgenic method were listed as experimental procedures. In the article of ref. 73, 10 micrograms of diethylstilbestrol daily on days 1 to 5 after birth were administrated for neonatal estrogenization. Are there any recommendations for estrogenization regimens in this research field? Please describe other papers' regimens.
-Accurate evaluation and grading of prostatic inflammation consider being significant points for these kinds of studies. Please describe commonly used methods to quantify prostatic inflammation.
References
-Published information (volume and page ranges) was written as "Epub ahead of print" in some references. Precise published information should be described if a paper has been published on printed media.
Author Response
Reviewer 2
Comments and Suggestions for Authors
Thank you very much for giving me this opportunity to review the article entitled "The translational role of animal models for estrogen-related functional bladder outlet obstruction and prostatic inflammation."
The authors overviewed sex hormone-related LUTS and its animal models clearly and concisely; however, some minor modifications are required.
The followings are comments and questions.
-Third part, Preclinical models for ---, seems to be the main point of this article. For the better understanding by readers who are interested in LUTS and animal experiments, it would be better to have a more detailed explanation. Some questions are listed below. Are there any recommendations for estrogenization regimens in this research field? Please describe other papers' regimens.
Neonatal estrogenization method and AROM transgenic method were listed as experimental procedures. In the article of ref. 73, 10 micrograms of diethylstilbestrol daily on days 1 to 5 after birth were administrated for neonatal estrogenization.
Reply: Thank you for addressing this very important practical issue. While establishing these hormonally induced disease models, we conducted an extensive literature review to understand what kind of hormone regimens have been used and importantly what results have been obtained. However, it is very difficult to make any recommendations about detailed regimens as several factors can influence the outcome of a treatment regimen, as observed in the large differences between the study outcomes in the literature we have reviewed. For example, “estrogenization” may be a consequence of different estrogen dosing routes, and in the adult rodent models, differences between the devices used to administer the hormones. We also highlight the differing hormonal sensitivity between certain strains of mice and rats. Furthermore, many studies have highlighted importance of the estrogen to androgen ratio. We added sentences regarding this issue to manuscript.
Added sentences to the manuscript:
Chapter 3: Several different approaches have been used to induce hormonal imbalance during the neonatal period or in adult mice and rats. Detailed experimental methods have not been described in this review but it is essential to understand that mice and rats, and different strains thereof, differ in their sensitivity to hormone imbalance. Therefore, it is always important to carefully follow the exposure methods used in the original publications.
Chapter 3.2: Although neonatal estrogenization in mice causes obstruction, it has also been observed to inhibit the growth of the prostate lobes and cause severe inflammation [57]. Thus, neonatal estrogenization was not considered an appropriate method for generating a representative model for prostatic inflammation studies in adult mice.
-Accurate evaluation and grading of prostatic inflammation consider being significant points for these kinds of studies. Please describe commonly used methods to quantify prostatic inflammation.
Reviewer 3 Report
Santi et al. propose a review regarding the role of animal models to study the effects of estrogens in the development of functional bladder obstruction and prostate inflammation. As presented in the manuscript, the role of the estrogens has even documented for any years, together with the prominent activity of androgens, specially DHT. This review has clearly some merit and focuses on an interesting matter: how could animal models be useful to screen new molecules for treating patients. Several points should however be clarified in order to increase the quality of this manuscript.
- Once again, even though the role of estrogens has been evoked for the development of LUTS, it is clear that the main pharmacological targets encompass 5-alpha reductase inhibitor (targeting thus the levels of DHT), anticholinergic compounds to treat hyperactive bladder, alpha-1 adrenergic antagonist or beta-3 adrenergic agonists to decrease the contraction of the smooth muscles surrounding urethra, or phosphodiesterase-5 (PDE-5) inhibitor. Altogether, it is puzzling that ER-blockers (SERMs or SERDs) or aromatase inhibitors are not used to treat LUTS, suggesting that finally estrogens have lower if not no role in the molecular mechanism leading to LUTS in human. Hence, the authors should described the main possible pharmacological treatments and comment this important point in the manuscript.
- The authors should indicate how their interesting animal models reacts after having been treated by the classical treatments, if this point has been investigated.
- Conversely, the role of estrogens in BOO is much clearer and the data presented by the authors are clear cut. One point needs some clarification: ventral prostate in mouse has been classically associated to the transition zone in humans, whose proliferation of epithelial and smooth muscle cells cause BPH. Conversely, dorsolateral prostate is usually compared to the peripheral and central zones in human, where >90% of the adenocarcinoma occur. How could this point fit with the development of functional BOO?
Author Response
Reviewer 3
Comments and Suggestions for Authors
Santti et al. propose a review regarding the role of animal models to study the effects of estrogens in the development of functional bladder obstruction and prostate inflammation. As presented in the manuscript, the role of the estrogens has even documented for any years, together with the prominent activity of androgens, specially DHT. This review has clearly some merit and focuses on an interesting matter: how could animal models be useful to screen new molecules for treating patients. Several points should however be clarified in order to increase the quality of this manuscript.
Once again, even though the role of estrogens has been evoked for the development of LUTS, it is clear that the main pharmacological targets encompass 5-alpha reductase inhibitor (targeting thus the levels of DHT), anticholinergic compounds to treat hyperactive bladder, alpha-1 adrenergic antagonist or beta-3 adrenergic agonists to decrease the contraction of the smooth muscles surrounding urethra, or phosphodiesterase-5 (PDE-5) inhibitor. Altogether, it is puzzling that ER-blockers (SERMs or SERDs) or aromatase inhibitors are not used to treat LUTS, suggesting that finally estrogens have lower if not no role in the molecular mechanism leading to LUTS in human. Hence, the authors should describe the main possible pharmacological treatments and comment this important point in the manuscript.
Reply: Thank you for these important considerations. As we aimed to study the role of the hormonal imbalance in the development of BOO and prostate inflammation, our pharmacological focus has mainly been based on novel hormonal treatment options. While preparing this manuscript we needed to limit the length of the review and therefore excluded any descriptions of pharmacological treatments in the current clinical use. Considering translational aspects when dealing with 5-alpha reductase inhibitors (which target the levels of DHT), there has not been to our knowledge any studies used in the animal models we have discussed. Also, our main purpose was to highlight the functional and urodynamic aspects of the models we discuss. 5-alpha reductase inhibitors are mainly used in BPH; BPH was not the main focus of the current manuscript. We have not found, any signs of hyperactive bladder in the models discussed, therefore anticholinergic compounds have not been used to treat hyperactive bladder. Additionally, in the accompanying estrogen-induced disease models, these compounds have not been used in the models we discuss. Although none of the clinical treatment options that are currently used have been tested in the animal models mentioned in this manuscript, we hope that future studies will focus on this.
To date, the treatment options available for patients with chronic prostatitis/chronic pelvic pain syndrome have provided moderate benefit to patients. The patient group is very heterogeneous and thus the clinical trial design for evaluating novel therapeutic options should be revised, a point discussed by Nickel JC, BJU Int. 2020. Also, to date, effects of hormonal imbalance in the male lower urinary tract function have been studied rather infrequently, and as such proper clinical trials to evaluate e.g. SERMs or SERDs have not yet been conducted. We believe that more information is needed to define the patient groups that may benefit from hormonal therapies. Indeed, Dr. Santti made careful and extensive planning for such a clinical trial to study the possible effects of SERMs in men with LUTS, but concluded that more information is needed to define the right patient group. During the manuscript preparation we carefully considered if more discussion should be addressed to these clinical considerations, but we finally made the decision to focus on preclinical models.
- The authors should indicate how their interesting animal models reacts after having been treated by the classical treatments, if this point has been investigated.
Reply: Thank you for addressing this very important point. As we have focused on hormone-induced disease, we have not considered testing of classical agents that have different mechanisms of action. We have tested several novel drug candidates and while some of our studies remain confidential and unpublished, other studies have been published eg. Konkol et al., 2017 Galactoglucomannan-rich hemicellulose extract from Norway spruce (Picea abies) exerts beneficial effects on chronic prostatic inflammation and lower urinary tract symptoms in vivo and Bernoulli et al., 2014 Effects of afala and antiestrogen ICI 182,780 in the model of hormone-dependent prostate inflammation. As the mechanisms of action in these papers were not directly hormone focused, we opted not include them in the manuscript.
Conversely, the role of estrogens in BOO is much clearer and the data presented by the authors are clear cut. One point needs some clarification: ventral prostate in mouse has been classically associated to the transition zone in humans, whose proliferation of epithelial and smooth muscle cells cause BPH. Conversely, dorsolateral prostate is usually compared to the peripheral and central zones in human, where >90% of the adenocarcinoma occur. How could this point fit with the development of functional BOO?
Reply: We agree that the identification of the rodent prostate lobes and their association with corresponding zones in humans is important. In our studies we have carefully quantified prostatic inflammation in the dorsolateral lobes during different hormonal exposure periods. We observed that the inflammation developed gradually from mild stromal inflammation into a more extensive and complex form. Longer hormonal treatment periods caused concomitant glandular acute inflammation and obstructive voiding. Precancerous PIN-like lesions and adenocarcinomas were observed in the very specific periurethal area after obstructive voiding developed. Thus, this model does not support the role of the adenocarcinoma in the development of BOO, since cancer developed in this model only after obstruction. We thank the reviewer for pointing out this important issue. In general, we knowingly excluded broader discussion relating to cancer from the scope of this article.
This manuscript is a resubmission of an earlier submission. The following is a list of the peer review reports and author responses from that submission.